# Cockle Shell-Derived Aragonite CaCO_3_ Nanoparticles for Co-Delivery of Doxorubicin and Thymoquinone Eliminates Cancer Stem Cells

**DOI:** 10.3390/ijms21051900

**Published:** 2020-03-10

**Authors:** Kehinde Muibat Ibiyeye, Abu Bakar Zakaria Zuki

**Affiliations:** 1Laboratory of Molecular Biomedicine, Institute of Bioscience, Universiti Putra Malaysia, Selangor 43400, Malaysia; kehindebiyeye@gmail.com; 2Department of Veterinary Preclinical Sciences, Faculty of Veterinary Medicine, Universiti Putra Malaysia, Selangor 43400, Malaysia

**Keywords:** cancer stem cell, breast cancer, nanoparticle, doxorubicin, thymoquinone

## Abstract

Cancer stem cells CSCs (tumour-initiating cells) are responsible for cancer metastasis and recurrence associated with resistance to conventional chemotherapy. This study generated MBA MD231 3D cancer stem cells enriched spheroids in serum-free conditions and evaluated the influence of combined doxorubicin/thymoquinone-loaded cockle-shell-derived aragonite calcium carbonate nanoparticles. Single loaded drugs and free drugs were also evaluated. WST assay, sphere forming assay, ALDH activity analysis, Surface marker of CD44 and CD24 expression, apoptosis with Annexin V-PI kit, cell cycle analysis, morphological changes using a phase contrast light microscope, scanning electron microscopy, invasion assay and migration assay were carried out; The combination therapy showed enhanced apoptosis, reduction in ALDH activity and expression of CD44 and CD24 surface maker, reduction in cellular migration and invasion, inhibition of 3D sphere formation when compared to the free drugs and the single drug-loaded nanoparticle. Scanning electron microscopy showed poor spheroid formation, cell membrane blebbing, presence of cell shrinkage, distortion in the spheroid architecture; and the results from this study showed that combined drug-loaded cockle-shell-derived aragonite calcium carbonate nanoparticles can efficiently destroy the breast CSCs compared to single drug-loaded nanoparticle and a simple mixture of doxorubicin and thymoquinone.

## 1. Introduction

Cancer recurrence and distant metastasis are the major causes of poor prognosis in cancer patients. These are related to a small fraction of stem-like cells termed as cancer stem cells (CSCs) or tumour-initiating cells (TICs) that exhibit an aggressive phenotype [1]. Breast cancer stem cells (BCSCs) have been isolated from breast cancer cell lines and breast samples from cancer patients based on a CD44^high/+^ CD24^low/−^ phenotype and/or high ALDH (aldehyde dehydrogenase) activity [2,3]. BCSCs play an essential role in tumorigenicity as well as local invasion and migration. BCSCs have the capacity for self-renewal, chemo and/or radio-resistance and are responsible for the initiation, progression and aggressiveness of distant metastatic lesion [4,5]. CSCs’ drug resistance is attributed to aldehyde dehydrogenase activity, epithelial-to-mesenchymal transition, aberrant ABC transporter activity or expression and enhanced DNA damage response [6]. The development of therapeutic approaches designed to target BCSCs may ultimately result in finding a cure for breast cancer.

High ALDH1 activity is correlated with poor prognosis in breast cancer patients and has been associated with chemo/radio-resistance [7]. ALDH is involved in the oxidation of intracellular aldehydes and retinol to retinoic acid during stem cell differentiation. ALDH1 accelerates detoxification by reducing ROS levels but also generating nicotinamide adenine dinucleotide phosphate, NADP, an antioxidant. However, ALDH has been shown to play a vital role in CSCs self-protection and resistance to chemotherapeutic drugs cyclophosphamide and paclitaxel [8].

CD44 is a transmembrane glycoprotein that has high specificity for hyaluronic acid and also interacts with osteopontin, collagen, laminin and matrix metalloproteinases. CD44 is responsible for control of numerous cellular functions, including cell adhesion, migration, homing, and transmission of growth signals [4,5,9]. Recent findings have shown that about 70% of breast cancer patients with early bone metastasis have tumours enriched in the CD44^+^ phenotype [3]. However, CD24 is poorly expressed in various cancer types and it is involved in cell adhesion and metastasis. CSCs and progenitor cells have the CD24^low/−^ phenotype [10].

Cancer cell invasion of the surrounding normal tissues is largely considered to be one of the key hallmarks of malignancy. The invasiveness and distant metastasis of cancer cells are responsible for the majority of cancer deaths, and cancer relapse that often presents with a more aggressive phenotype [11,12].

A few therapeutic agents have been shown to be cytotoxic to CSCs including salinomycin [13], curcumin [14], disulfiram [15] and chloroquine [16]. However, poor drug solubility and stability, system side effects, a short systemic half-life, poor biodistribution and low therapeutic index limit the efficacy of these drugs in vivo [17].

In designing anticancer drugs for the delivery of drugs specific to cancer cells including cancer stem cells, the ability of the delivery system to sustain a high drug concentration in the cancer environment, improve anti-cancer efficacy and reduce systemic side effects is important. The efficacy of commonly available anticancer drugs is limited by systemic side effects and multidrug resistance. On the other hand, significant progress has been made in nanotechnology to overcome these problems [18].

Nanoparticles are an excellent carrier for the delivery of multiple drugs. It is expected that encapsulating two or more drugs into a single nanoparticle will enable the release of the drugs at balanced rates and ratios [19]. Combining anticancer drugs with different mechanisms of action will result in additive or synergistic effects, thereby reducing multidrug resistance [20,21]. Nano-sized carriers have shown noteworthy potential in CSC-targeted drug delivery [22].

Cockle-shell-derived aragonite calcium carbonate nanoparticles (ACNP) have exhibited promising potential as targeting nano-sized carriers against cancer cells [23,24,25,26]. The pH-sensitive drug release of ACNP provides a possibility of targeting cancer cells as well as CSCs and controlled delivery of anticancer drugs.

In this study, we determine the sensitivity of doxorubicin/thymoquinone-loaded ACNP as well as single-loaded drugs, free doxorubicin and thymoquinone on BCSCs. Sphere formation assay, morphological assessment, apoptotic assay as well as cell cycle analysis were assessed.

A 3D tumoursphere model is commonly used to evaluate CSCs proliferation, migration, invasion, angiogenesis, immune interactions as well as drug screening and development of novel therapeutic agents [27,28]. Commonly, hanging drop assay, ultra-low attachment plates, and microfluidic devices are used to generate the tumoursphere. A 3D tumoursphere cell culture is largely acknowledged to be comparable to the tumour microenvironment in vivo. In addition, a 3D tumoursphere culture could reduce the reliance on animal models for drug screening [28].

## 2. Results and Discussion

### 2.1. Characterization of MDA MB 231 3D Mammospheres-Enriched Cancer Stem Cells

The 3D mammospheres enriched with BCSCs were generated by culturing MDA MB 231 in ultra-low attachment plates for 7 days, as previously described with slight modifications [29]. The light microscopy images showed a grape-like appearance of the mammosphere formed by MDA MB 231 with no obvious changes in the morphology of the 3D mammospheres after the three serial passages (Figure 1). A similar morphology was reported by Eguchi et al. [30] and Wang et al. [29]. Eguchi et al. cultured 67 types of cell lines in ultra-low attachment plates or NanoCulture Plates, eight cell lines formed a grape-like aggregation, including MDA MB 231. Among these eight cell lines, six were derivative of adenocarcinoma. MDA MB 231 is human breast adenocarcinoma therefore, cannot form a perfect sphere or spheroid [30].

The proficiency for self-renewal of the 3D mammosphere cells was demonstrated by their ability to form almost the same number and morphology of mammospheres after serial passaging in vitro. It has also been shown that as few as 5000 mammosphere cells can develop into orthotopic xenograft breast tumours in mice, whereas the same quantity of parent cancer cells could not [31].

The resulting mammospheres are not just aggregation of MDA MB 231 cells cancer cells, but enriched with CSCs (Figure 2). Suspension culture in an ultra-low attachment plate promotes enrichment of CSCs. Other cells will die by anoikis—programmed cell death due to loss of contact.

To further study the stem properties of the mammosphere as related to parental MDA MB 231 cells, the expression of CD44, CD24 and Aldehyde dehydrogenase activity ALDH were studied with flow cytometry. CD44^+^CD24^−/low^ receptor expression is widely used to identify CSCs in breast cancers. CD44 is a glycoprotein found on the cell surface. It is involved in cell to cell interactions, cell adhesion, and migration. CD24 is also a cell adhesion molecule. Up-regulation of CD24 inhibits stemness in breast cancer cells and its reduced expression is linked to breast cancer aggressiveness. The ALDH superfamily of enzymes is involved in detoxification of intracellular aldehydes and overexpression of ALDH1 has been shown to be responsible for chemoresistance. Interestingly, CD44, CD24 and ALDH sets of markers identify overlapping but not identical cell populations [32].

The current data shows that CD44^+^CD24^−/low^ is enriched in 3D mammospheres cells (92%) compared to parental cells (70%) (Figure 3). Jafari et al. [33] reported the percentage of CD44^+^CD24^−/low^ in parental MDA MB 231 cells to be 73.2% and 85.1% in a 3D mammosphere. Croker et al. [34] also report that the MDA MB 231 cell line contains 79.5 % CD44^+^CD24^−/low^. MDA MB 231 is a triple-negative cell line and highly metastatic. Therefore, it is not unusual to have a high level of CD44^+^CD24^−/low^ cells in the MDA MB 231 parental cells when compared to other breast cancer cell lines; less than 1% in both parental MCF 7 and MDA-MB-468 [34,35]. Additionally, the expression of CD24 in the 3D mammosphere cells was noticed to be low (6%), on the other hand, it is very much expressed in the parental MDA MB 231 cells. The rate of expression of CD44^+^CD24^−/low^ is more or less the same as that of CD44^+^ cells in mammosphere cells (Figure 3). ALDH activity was 0.25% in 3D mammosphere cells (Figure 4). Marcato et al. [36] also reported ADLH activity in MDA MB 231 to be less than 1%.

### 2.2. Drug Sensitivity Assays

The effects of drug loaded on mammosphere cells in 3D mammospheres, single mammosphere cells (cells that are dissociated from mammospheres) and parent MDA MB 231 cells (2D model) were investigated in this study. The cytotoxic 3D mammospheres and single mammosphere cells were cultured in the ultra-low attachment plate, while parent MDA MB 231 was cultured as a monolayer in conventional culture plates. The 3D and 2D models were studied for comparison because both models are typically used to study cancer stem cells in the cancer research literature. We tested free Dox versus Dox-ACNP, free TQ versus TQ-ACNP and free Dox/TQ versus Dox/TQ-ACNP (DOX: TQ = 3:2) by incubating them with the cells at 0 to 100 µg/mL for 10 days. Samples treated with blank ACNP and with no treatment were also studied as controls. The 10 days treatment duration is used to justify the efficacy of the drugs in preventing possible cancer cell reactivation after treatment in vitro [31].

As shown in Figure 5, cells in the mammosphere, 3D mammosphere and single cell in 3D, including cancer stem cells, are sensitive to Dox-ANCP and Dox/TQ-ACNP and more resistant to free Dox and Dox/TQ. Cancer stem cells were known to be resistant to most chemotherapy, this could be effectively overcome by treating the mammosphere cells with Dox-ACNP and also using combination therapy of Dox/TQ-loaded ACNP. Interestingly, free TQ was more sensitive to mammosphere cells. TQ-ACNP is the least sensitive in all three models. This may be due to a delayed release of TQ from ACNP, which decreases the bioavailability of the hydrophobic drug during incubation.

For cells in a monolayer, at a lower dose (3.125 to 12.5 ug/mL), the cells were more sensitive to Dox-ACNPs compared to free Dox; as the dose increased, there were no significant differences in the cell viability. Cells in the monolayer are more sensitive to free TQ compared to TQ-ACNP but free Dox/TQ and Dox/TQ-ACNP showed no significant differences in the cell viability in all the doses. Interestingly, the cell viability of 2D or cells in a monolayer for free drug (except for TQ) is higher than the loaded ACNPs after 10 days, which is in contrast to what was observed when the treatment was for 24, 48 and 72 h [25]. Although the duration of treatment and doses were not similar, the cell viability progressively reduce as the treatment dose increased in a time- and dose-dependent manner. At 10 days, the cell viability of MDA MB 231, after treatment with Dox-ACNP, progressively decreased from 21.2% to 13.2% at a treatment dose of 3.125 µg/mL and 100 µg/mL, respectively; for free Dox, from 46.0% to 14.2%, respectively. For TQ-ACNP, from 76.9% to 41.9%, respectively; for free TQ from 76.9% to 13.6%, respectively. For Dox/TQ-ACNP, from 35.1% to 15.2%, respectively; for Dox/TQ from 37.6% to 14.6%, respectively.

For cell viability of single cell 3D, Dox-ACNP at 3.125 µg/mL and 100 µg/mL yielded 36.1% and 16.2%, respectively; free Dox yielded 52.7% and 38.7%, respectively. For TQ-ACNP, cell viability was 97.7% and 38.6%, respectively; free TQ 42.1% and 37.1%, respectively. The cell viability of Dox/TQ-ACNP and free Dox/TQ decreased progressively from 26.3% to 11% and 55.2% to 37.3%, respectively.

For the 3D model, the viability for Dox-ACNP was 73.1% and 13% at 3.125 µg/mL and 100 µg/mL, respectively; for free Dox, it was 40.6% and 28.2%, respectively. For TQ-ACNP, it was 85% and 48.2%, respectively; for free TQ, it was 73.9% and 26.8%, respectively. For Dox/TQ-ACNP and free Dox/TQ, the cell viability decreased progressively from 34.2% to 9.6% and 45.7% to 26%, respectively.

Sarisozen et al. [37] also reported similar findings with doxorubicin–curcumin PEG–PE-based polymeric micelles nanoformulations on glioblastoma 3D spheroid cells. Increased efficacy of combination PEG–PE-based polymeric micelles nanoformulations was observed on the glioblastoma spheroids compared to a single drug-loaded nanoformulation [37]. Doxorubicin-tethered gold nanoparticles potentiate drug delivery to breast CSCs, diminishes mammosphere formation and CSCs cancer initiation capacity, while doxorubicin alone increased the enrichment of breast CSCs [38]. The doxorubicin and irinotecan-loaded hyaluronan-decorated nanoparticles destroyed the CSCs about 500 times more efficiently compared to the free combination of the two drugs in both in vitro and in vivo studies [29].

### 2.3. IC_50_ and Combination Index

The IC_50_ (inhibitory concentration at 50% viability) was quantified with Graphpad Prism 7 by using the cell viability data shown in Figure 5. As shown in Figure 6 and Table 1, the IC_50_ of free Dox in 3D mammosphere cells (5.29 μg/mL, ∼2 times higher) was significantly higher than that of the single cells 3D (2.339 μg/mL) and cells in monolayer (2.705 μg/mL, 2 times lower). This shows the resistance of 3D mammosphere cells to free Dox compared to the cells in monolayer. The IC_50_ of Dox-ACNP in 3D mammosphere (4.133 μg/mL) was not significantly higher than that of single cells 3D (2.342 μg/mL) and cells in monolayer (2.227 μg/mL), but it is slightly lower than that of free Dox (5.29 μg/mL).

Interestingly, the IC_50_ of free TQ in 3D mammosphere cells (7.142 μg/mL, ∼3 times higher) was significantly higher than that of the single cells 3D (2.175 μg/mL,) and significantly lower than cells in the monolayer (16.95 μg/mL, ∼2 times lower). This shows that mammosphere cells including CSCs are more sensitive to free TQ. Although the IC_50_ of TQ-ACNP in mammosphere cells (3D and single cell 3D, 16 μg/mL and 15.65 μg/mL, respectively) are statistically significant as compared to cells in monolayer (14.1 μg/mL), their IC_50_ values are ∼1 μg/mL different from one another. The slight difference in the IC_50_ values of ACNP may be due to delayed release of TQ from ACNP, as aforementioned above.

There is no significant difference in the IC_50_ values of Dox/TQ-ACNP in the 3D mammosphere, single cell 3D and cells in monolayer. Therefore, Dox/TQ-ACNP is able to kill both cancer stem cells and the bulk of cancer cells. For 3D mammosphere cells, the IC_50_ of Dox/TQ-ACNP is 2.446 μg/mL, which is ~2 times lower than both free Dox and Dox-ACNP, and ~3 times, ~6 times lower than TQ and TQ-ACNP, respectively. The results showed that the combined drugs-loaded cockle-shell-derived aragonite calcium carbonate nanoparticles can efficiently destroy the breast CSCs as compared to a single drug-loaded nanoparticle.

The doses of Dox-ACNP and Dox at 3.2 μg/mL, TQ-ACNP and TQ at 10 μg/mL, Dox/TQ-ACNP and Dox/TQ at 2.4 μg/mL were used for further experiments to compare the free and ACNP-loaded counterpart; i.e., the average of the IC_50_ value of Dox and TQ for the 3D mammosphere model.

The combination index (CI) along with results interpretations were calculated for 3D mammosphere model (Table 2). Dox/TQ-ACNP combination treatment exhibited synergism in the 3D mammosphere model, with free Dox/TQ treatment exhibiting antagonism.

### 2.4. Apoptosis

The induction of apoptosis in the 3D mammosphere with free and drug-loaded ACNP was investigated using Annexin V/PI flow cytometry analysis. Annexin V-FITC stains cells at an early stage of apoptosis, while PI enters the cells through the damaged cell membrane and bind with DNA of necrotic cells.

As shown in Figure 7 and Figure 8, for the Dox-ACNP treatment group, the percentages of early apoptotic cells at days 3 and 10 were 5.3% and 4.4%, respectively; for free Dox, they were 7.6% and 18.1%, respectively. The percentages of late apoptotic cells for Dox-ACNP at days 3 and 10 were 40.7% and 44.8%, respectively; for free Dox, they were 17.9% and 35.1%, respectively. The percentage of necrotic cells for Dox-ACNP were 35.5% and 20.5%, respectively; for free Dox, they were 6.6% and 4.6%, respectively. At a dose of 3.2 µg/mL, Dox-ACNP is more effective in inducing apoptosis in the 3D mammosphere compared to free Dox.

A dose of 10 µg/mL was used to compare the induction of apoptosis in the 3D mammosphere treated with free TQ and TQ-ACNP. For TQ-ACNP, the percentages of early apoptotic cells at days 3 and 10 were 7.4% and 9.6%, respectively, while for TQ, they were 9.2% and 57.2%, respectively. The percentages of late apoptotic cells for TQ-ACNP at days 3 and 10 were 10.5% and 13.5%, respectively, while for TQ, they were 52.4% and 9.6%, respectively. The percentage of necrotic cells for TQ-ACNP were 2.4% and 4.4%, respectively, while for TQ, they were 26.5% and 3.2%, respectively. TQ is more effective in inducing apoptosis in the 3D mammosphere when compared with TQ-ACNP. This may be due to the delayed release of loaded TQ from TQ-ACNP.

For Dox/TQ-ACNP, the percentages of early apoptotic cells at days 3 and 10 were 3.3% and 9.9%, respectively, while for Dox/TQ, they were 5.9% and 17.3%, respectively. The percentages of late apoptotic cells for Dox/TQ-ACNP at days 3 and 10 were 20% and 36.3%, respectively, while for Dox/TQ, they were 20% and 36.9%, respectively. The percentage of necrotic cells for Dox/TQ-ACNP were 32.9% and 12.9%, respectively, while for Dox/TQ, they were 8% and 4.9%, respectively. At a dose of 2.4 µg/mL, which is less than 3.5 µg/mL for Dox-ACNP and 10 µg/mL for TQ-ACNP, combined drugs-loaded ACNP is more effective in inducing apoptosis in the CSCs compared to single drug-loaded ACNP.

### 2.5. Morphology Assessment

Grape-shaped mammospheres were observed in control for both single cell 3D and 3D mammosphere models (Figure 9). A similar morphology was observed in the samples treated with blank ACNP, signifying that ACNP is non-toxic to mammosphere cells comprising CSCs. Interestingly, TQ, TQ-ACNP and Dox/TQ were observed to inhibit the development of mammospheres by CSCs in the single cell 3D model; no grape-shaped mammosphere was noticeable at day 10. Smaller mammospheres were observed in single-cell 3D samples treated with Dox, Dox-ACNP and Dox/TQ-ACNP. At day 10, 3D mammospheres were noticed to have reduced in size for most treatments, except for TQ, with Dox/TQ and Dox/TQ-ACNP treatments having the smallest size. TQ-treated 3D mammosphere cells are no longer aggregated but were separated as individual cells. The combination of TQ and Dox effectively reduced the size of the mammosphere at a dose lower than the single Dox or TQ treatment.

The surface morphology of the 3D mammosphere was observed with SEM (Figure 10). The control sample showed a well formed mammosphere with cells exhibiting a smooth surface and some showing membrane blebbing; a sign of early apoptosis. These cells are likely undergoing anoikis; only CSCs are resistant to anoikis. No other abnormalities were seen. Treated cells showed cell shrinkage, loss of membrane integrity, and membrane blebbing. Dox/TQ-ACNP-treated mammosphere showed a more abnormal surface morphology of the mammosphere cells with abnormal mammosphere formation compared to the other treatment. The mammosphere in the TQ-treated sample was loose—not as compacted as the control. This is similar to the findings in the light microscopy image.

### 2.6. Sphere-Forming or Self-Renewal Efficiency

The aim of this experiment was to determine if drug-loaded ACNP could selectively target CSCs. When breast cancer cells are cultured in CSC media, they grow and form 3D mammospheres that are considerably enriched in CSCs and early undifferentiated progenitor cells. Only self-renewing CSCs in 3D mammosphere cells that are dissociated into single cells and then cultured in MammoCult media, can form a sizeable mammosphere in succeeding passages (secondary/tertiary passage) but not progenitor cells. Compounds that reduce proliferation or growth in primary and also succeeding passages in 3D culture without additional treatment, suggest inhibition of self-renewal, a characteristic that only CSCs possess but not early undifferentiated progenitor cells [39].

Figure 11 shows the number of mammospheres formed after various treatments at first, second and third passage. Inhibition of sphere formation was noticed in TQ, TQ-ACNP, Dox/TQ and Dox/TQ-ACNP. The TQ-treated sample did not show any mammosphere formation from the first passage to the third passage, while TQ-ACNP, Dox/TQ and Dox/TQ-ACNP formed 2, 2 and 3 mammospheres per 4000 cells seeded, respectively. Therefore, based on the number and size of mammosphere formed, TQ and Dox/TQ-ACNP efficiently inhibit mammosphere formation, thereby inhibiting the self-renewal capacity of CSCs.

### 2.7. Surface Marker of CD44 and CD24

The most common method to identify BCSCs is by studying the expression of distinctive cell surface markers. CD44^high/+^CD24^low/−^ is one of such markers [40]. The CD44^high/+^CD24^low/−^ population of cells are 1000 times more tumorigenic than the CD44^low/−^CD24^high/+^ or CD44^high/+^CD24^high/+^ cell population; as few as 200 CD44^high/+^CD24^low/−^ cells leads to tumour formation after injection in SCID mice. CD44^high/+^CD24^low/−^ have been shown to be involved in cancer invasion and metastasis [41,42].

The effect of free and drug-loaded ACNP on stem cell surface marker CD44^+^CD24^−^ was assessed using flow cytometry. As shown in Figure 12 and Figure 13, decreased expression of the surface marker was noticed for all treatments at days 3 and 10 as compared to control 84.3%; Dox 44.05% and 51.4%, respectively; Dox-ACNP 56.15% and 79.55%, respectively; TQ 76.65% and 81.4%, respectively; TQ-ACNP 63.85% and 67.2%, respectively; Dox/TQ 37.3% and 42.5%, respectively; Dox/TQ-ACNP 6.2% and 19.4%, respectively. Dox/TQ-ACNP suppressed the expression of the cancer stem cell surface marker the most at both days 3 and 10 (6.2% and 19.4%, respectively).

Interestingly, TQ eliminates CSCs (inhibition of self-renewal capacity, Figure 11) but spares the early undifferentiated progenitor cells (slight reduction in the expression of CD44^+^CD24^−^ compared to control). As previously mentioned, both CSCs and early undifferentiated progenitor cells express CD44^+^CD24^−^. The combination of Dox/TQ-ACNP effectively eliminates both CSCs and the undifferentiated progenitor cells.

### 2.8. ALDH Activity

A corresponding approach for identifying CSCs involves measuring ALDH activity. ALDH is involved in the oxidation of intracellular aldehydes. ALDH1 facilitates detoxification by reducing reactive oxygen species (ROS) levels. The inability to manage ROS load leads to oxidative stress, lipid peroxidation which generates highly reactive and toxic aldehydes. In CSCs, ALDH activity abrogates oxidative stress and imparts resistance against chemotherapeutic agents [8,43] and has been associated with poor clinical outcome in patients with inflammatory breast cancer [7].

It is, therefore, not surprising to observe an increase in the ALDH activity for all treated samples at day 3 (Figure 14). The increase is because the CSCs are detoxifying the drugs to prevent oxidative stress. The most increase ALDH activity was noticed in treatments that had Dox (Dox 11.7%, Dox-ACNP 3.3%, Dox/TQ 9.05%, and Dox/TQ-ACNP 14.6% at day 3. TQ- and TQ-ACNP-treated samples showed the lowest ADLH activities at day 3; 1.5% and 0.75%, respectively. TQ is a phytochemical compound that has a wide safety margin in both in vivo and in vitro studies [44]. There was no significant difference in ADLH activity for TQ and TQ-ACNP at day 3 and 10.

At day 10, Dox/TQ-ACNP drastically reduced ALDH activity to zero percent, as well as the percentage of viable cells (0.1%) (Figure 14 and Figure 15). Dox/TQ-ACNP reduced ADLH activity as well as induced cytotoxicity against CSCs, and thereby overcomes CSCs drug resistance. Decrease in ALDH activity was also noticed in other treatments (Dox 3.95%, Dox-ACNP 1.95%, TQ 2.05%, TQ-ACNP 0% and Dox/TQ 1.1% at day 10). The reduction of ADLH activity in Dox, Dox-ACNP and Dox/TQ may be due to successive efflux of these drugs from the CSCs.

### 2.9. Anti-Metastatic Effect

Invasion and metastasis is one of the important hallmarks of cancer. Cancer cells metastasis potential can be measured by wound healing and cell invasion assay. The percentage wound closure was highest in the TQ-ACNP (71.6%) group when compared to other treatment groups at 24 h. In ascending order, DoxTQ (0%), TQ (25.2%), Dox/TQ-ACNP (32.3%) demonstrated the least closure at 24 h (Figure 16). DoxTQ and TQ are free drug and their onset of action is quicker than Dox/TQ-ACNP. This may be why both reduced cell migration more effectively compared to Dox/TQ-ACNP. Apoptosis and cell detachment contributed to the impaired cell migration.

The percentage of cell invasion across the basement membrane is shown in Figure 17. TQ-ACNP (97.7%) had the least effect on the invasion of cells across the basement membrane, followed by Dox-ACNP (80.1%) and Dox (71.3%) when compared to control. Dox/TQ (41.8%) had the largest effect on cell invasion followed by Dox/TQ-ACNP (51.8%) and TQ (54%).

From the results of wound healing and cell migration assay, it is can be seen that TQ played a major role in the inhibition of CSCs migration and invasion; but the combination of Dox and TQ was more effective and Dox/TQ-ACNP was most effective among the loaded ACNP. Thymoquinone has been shown to inhibit cancer cell migration and invasion. TQ controlled melanoma metastasis [45], inhibited cell migration and invasion in human glioma cells by down-regulating focal adhesion kinase and inhibiting secretion of matrix metalloproteinase [46]. TQ also decreased the activity and expression of TWIST1 promoter, which resulted in the inhibition of breast cancer cell migration, invasion and metastasis induced by the epithelial–mesenchymal transition [47]. Therefore, free Dox/TQ and, most importantly, Dox/TQ-ACNP will prevent cancer invasiveness and distant metastasis, which is responsible for the majority of cancer deaths and cancer relapse.

### 2.10. Cell Cycle Analysis

To assess the influence of free and drug-loaded ACNP on the cell cycle distribution of breast CSCs, 3D mammosphere cells were treated with various treatments under investigation for 48 h and DNA content was also assessed using flow cytometry. All treatment were cell-cycle nonspecific.

No treatment caused a significant anti-proliferative effect by increasing the cell population at the G0/G1 phase after 48 h of treatment; Control 91.05%, Dox 70.1%, Dox-ACNP 69.65%, TQ 57.15%, TQ-ACNP 69.6%, Dox/TQ 75.7%, and Dox/TQ-ACNP 75.25% (Figure 18 and Figure 19).

All of the treatment caused significant S phase arrest by increasing its cell population from 7.8% (Control) to 12.15% (Dox), 13.4% (Dox-ACNP), 18.35% (TQ), 14.4% (TQ-ACNP), 9.15% (Dox/TQ) and 11.4% (Dox/TQ-ACNP). TQ and TQ-ACNP had the largest effect on S phase (18.35% and 14.4% increase in cell population, respectively), while Dox/TQ-ACNP and Dox/TQ had the least effect (11.4% and 9.15%, respectively).

The percentage of cells in the G2/M phase was increased in Dox, Dox-ACNP, TQ and Dox/TQ (13.6%, 12.3%, 14.55%, 11.7%, respectively) compared to control (9.8%), signifying cycle arrest in G2/M phase. After 48 h, all treatment options induced a significant increase in the subG0 cell population.

## 3. Materials and Methods

### 3.1. Preparation of ACNP and Drug Loading

The preparation of ACNPs, drug loading of thymoquinone (Tokyo chemical industry, Tokyo Japan) and doxorubicin (Cayman Chemical, Ann Arbor, MI, USA), characterization of Dox-ACNP, TQ-ACNP and Dox/TQ-ACNP were carried out in accordance with Ibiyeye et al. [48].

### 3.2. Generation of 3D Mammosphere from MDA MB 231

For mammosphere culture, MDA MB 231 cells (ATCC, Manassas, VA, USA) with less than ten passages were grown to about 70%–80% confluence and then enzymatically detached with trypsin (Nacalai Tesque, Kyoto, Japan), gently pipetted and suspended into single cells and passed through a 40 μm cell strainer (Corning, Lowell, MA, USA). A quantity of 1 mL of MDA MB 231 cells suspension was cultured in 24 well ultra-low attachment plates (Corning, Lowell, MA, USA) at 20,000 cells/mL in Mammocult complete media (Stem Cell Technologies, Vancouver, BC, Canada) supplemented with 4 μg/mL of heparin solution (Stem Cell Technologies, Vancouver, BC, Canada), 0.48 μg/mL hydrocortisone stock solution (Stem Cell Technologies, Vancouver, BC, Canada), and 1% antibiotics (Nacalai Tesque, Kyoto, Japan). After 7 days of culture in the incubator in 5% CO_2_ and at 37 °C, mammospheres were used for further experiment [29].

### 3.3. Drug Sensitivity Assays

For study of the CSCs drug sensitivity, the 3D mammospheres that were obtained after 7 days of culture in 24-well ultra-low attachment plates were then incubated for a further 10 days in fresh MammoCult complete media containing different concentrations of drug (ranging from 0 to 100 µg/mL). On day 10, WST-1 reagent (Roche, Mannheim, Germany) was added to each well and incubated for 4 h at 37 °C. The absorbance was read at 450 nm using a microplate reader (Tecan Infinite, Seestrasse, Mannedorf, Switzerland) to determine cell viability. Cell viability was calculated as the ratio of absorbance of each treatment group to control [29].

### 3.4. Combination Index (CI)

The CI was calculated using CompSyn software version 1.0 (ComboSyn Inc, Paramus, NJ, USA), to evaluate the synergism between the two drugs using the classic isobologram equation of Chou-Talalay. CI>1.3 antagonism; CI 1.1–1.3 moderate antagonism; CI 0.9–1.1 additive effect; CI 0.8–0.9 slight synergism; CI 0.4–0.8 synergism; CI 0.2–0.4 strong synergism [49].

### 3.5. Annexin V Assay

The effect of drug-loaded ACNPs apoptosis of CSCs was assessed using an Annexin V-PI kit (Nacalai Tesque, Kyoto, Japan). Annexin V has a strong affinity for phosphatidylserine. Phosphatidylserine diffuses from the inner to outer cell membrane in cells undergoing apoptosis. The phosphatidylserine can then be quantified with a flowcytometer by targeting it with FITC-tagged Annexin V. The procedure was performed according to the manufacturer’s instructions. Briefly, mammosphere cells were seeded (40,000 cells/well) in a 6-well ultra-low attachment plates followed by 24 h incubation. Ten days after treatment, the mammospheres were trypsinised into single cells, washed with PBS, and incubated in Annexin V-FITC/PI solution (5 uL Annexin V and 5 uL PI) for 15 min in the dark. The cells were then analysed with FACSCalibur Flow Cytometer (BD Biosciences, San Jose, CA, USA).

### 3.6. Morphology Assessment

#### 3.6.1. Light Microscopy Imaging

The morphological changes of mammospheres were captured and recorded using an inverted light microscope (Nikon, Minato City, Tokyo, Japan). Mammosphere images were taken at different intervals during the 10 days incubation period to measure the time-dependent size changes of the mammospheres for the different treatment groups and control.

#### 3.6.2. Scanning Electron Microscopy (SEM) Imaging of Mammospheres

Surface morphologies of MDA-MB-231 cells were observed using SEM. Day 7 mammospheres were treated with fresh MammoCult complete media containing different drug formulations and incubated for 72 h at 37 °C. The mammospheres were washed with PBS, fixed in 2.5% (*v*/*v*) glutaraldehyde and kept in 4 °C for 4 h. Then, they were washed with buffer three times and post-fixed with 2% osmium tetroxide for 2 h at 4 °C, dehydrated using different concentrations of ethanol (35%, 50%, 70%, 80%, 85% and 95%) for 10 min each and twice at 100%. Thereafter, the mammospheres were immersed in acetone for 10 min. The cell samples were mounted onto an aluminum stub, point dried, sputtered gold coated (E5100 Polaron, UK) and they were examined under SEM (JOEL-64000, Akishima, Tokyo, Japan). Surface and morphological changes in the cells are then observed and recorded [26].

### 3.7. Sphere Forming or Self-Renewal Efficiency

Sphere forming assay was performed as previously described by Patel et al. [39] with slight modification. MDA MB 231 cells with less than ten passages were grown to about 70%–80% confluence and then enzymatically detached with trypsin into single cell suspension and passed through a 40 μm cell strainer. A quantity of 2 mL of single cells suspension was then seeded at 20,000 cells per mL in a 6-well ultra-low attachment plate. The number of spheres in each well was assessed at day 7 after seeding and the sphere formation rate was calculated. Sphere-forming efficiency (SFE) for each passage was calculated as thus:(The number of spheres formed × 100)/The cells seeding density per well [first to the third generation](1)

### 3.8. Surface Marker of CD44 and CD24 by Flow Cytometry

To determine the percentage population of CSCs in the mammospheres, the expression of CD44^+^/CD24^−^ surface markers were measured. Day 7 mammospheres were treated with drug-loaded ANCP and free drugs. At days 3 and 10 of treatment, 3D mammospheres were transferred into centrifuge tubes, dissociated into a single cell suspension by trypinization and passed through a 40 μm filter. The cells were then stained with conjugated CD44 and CD24 antibodies (at 1:10 dilution) for 15 min at 4 °C, and washed with ice-cold PBS [31].

### 3.9. ALDH Activity Analysis by Flow Cytometry

Aldehyde (ALDH) activity was analysed using the Aldefluor kit (Stem Cell Technologies, Vancouver, BC, Canada) according to the manufacturer’s instructions. Briefly, Day 7 mammospheres were treated with drug-loaded ANCP and free drugs. At days 3 and 10 of treatment, 3D mammospheres were transferred into centrifuge tubes, dissociated into suspension of single cells and passed through a 40 μm filter. Activated Aldefluor reagent was added to the single cell suspension. A tube of cells with diethylaminobenzaldehyde (DEAB) specific inhibitor and another one without the inhibitor were used as controls and both were incubated at 37 °C for 45 min. After incubation, the test tubes and controls were centrifuged and the supernatant discarded, then resuspended in 0.5 mL Aldefluor buffer on ice and analysed with a FACSCalibur Flow Cytometer (BD Biosciences, San Jose, CA, USA).

### 3.10. Wound Healing Assay

3D-mammosphere-derived cells were dissociated into single cells and grown to confluence in a 6-well plate. Wound was initiated using a sterile 200 μL pipet tip through the cells, then cells were carefully rinsed with PBS. Images of the wound were captured using an inverted microscope (Nikon, Tokyo, Japan) at 6 and 24 h after. The areas of wound closure at different times were evaluated using Imagej software (NIH, Bethesda, Maryland, USA) to calculate the rate of cellular migration [35].

### 3.11. Cell Invasion Assay

Cell invasion assay was analysed using CHEMICON cell invasion assay kit (EMD Millipore, Burlington, MA, USA) according to their manufacturer’s instructions. Briefly, 3D mammospheres were dissociated into single cells, and cell suspension of 0.5 × 10^6^ cells/mL in serum free DMEM/F12 media. A quantity of 500 µL of media containing 10% FBS was added to the lower chamber, while 300 ul of cell suspension was added to the upper insert. After appropriate treatment, the invasion chamber with the cells was incubated for 48 h. At the end of the assay, the non-invading cells along with the matrix gel were removed with a cotton swab. The invading cells on the lower surface of the membrane were stained with staining solution for 20 min. The inserts were rinsed several times in PBS and allowed to air dry. For quantifying the invasive cells, stained cells were dissolved in 10% acetic acid—200 ul/well—and a constant amount of the dye/solute mixture was transferred to a 96-well plate for colorimetric reading at 560 nm. The percentages of invaded cells were calculated thus:(Absorbance of the sample × 100)/Absorbance of control(2)

### 3.12. Cell Cycle Analysis

The effect of drug-loaded ACNPs on the cell cycle was determined by using a propidium iodide flow cytometry kit (abcam, Cambridge, MA, USA). Mammospheres, enriched with BCSCs, were treated with drug-loaded ACNP and free drugs for 48 h. The mammospheres were then detached and dissociated into single cells. The single cells were fixed in ice-cold 66% ethanol and stored for at least 2 h at 4 °C. Cells were centrifuged at 1200 rpm for 5 min, supernatant was removed and then washed in PBS. The cells were resuspended in 200 uL of propidium iodide (PI) plus RNase staining solution and incubated at 37 °C in the dark for 30 min. Cells were placed on ice, in dark, and analyzed with FACSCalibur Flow Cytometer (BD Biosciences, San Jose, CA, USA).

### 3.13. Statistical Analysis

Graphpad Prism version 7 (GraphPad Software, San Diego, CA, USA) was used for all statistical analysis. The data were presented as statistical means ± S.E. The *p* value < 0.05 was set to be significant. The statistical comparisons were done using one-way or two-way ANOVA with Tukey post hoc test.

## 4. Conclusions

This study assessed the influence of combined doxorubicin/thymoquinone-loaded cockle-shell-derived aragonite calcium carbonate nanoparticles as well as single loaded drugs and free drugs on CSCs survival, self-renewal capacity, ALDH activity and expression of CD44^+^CD24^−^ surface maker and their metastatic potential. The combination therapy efficiently destroyed the breast CSCs and suppressed the CSCs properties and thus may be a potential curative strategy for the management of breast cancer recurrence and metastasis.

## Figures and Tables

**Figure 1 ijms-21-01900-f001:**
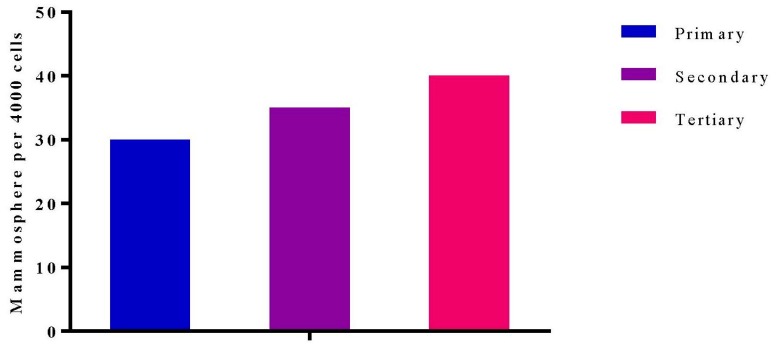
Graph shows there is no significant change in the number mammosphere formed at 3 different passages.

**Figure 2 ijms-21-01900-f002:**
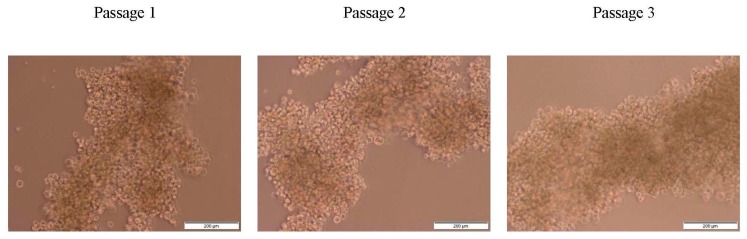
Microphotographs of mammospheres show no obvious changes in morphology at passage 1 to 2 cultured in ultra-low attachment plates.

**Figure 3 ijms-21-01900-f003:**
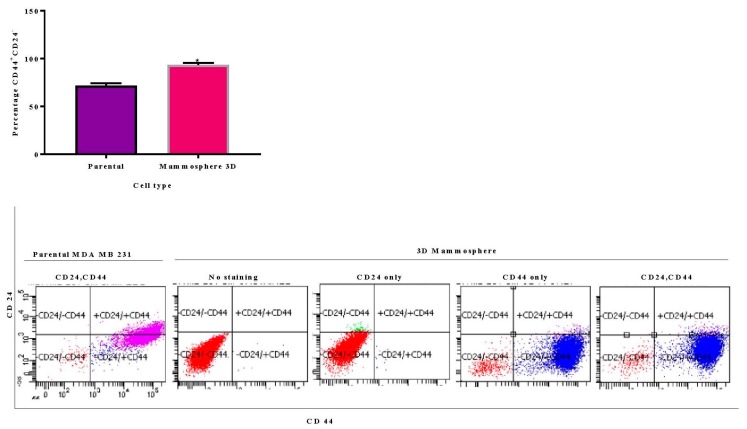
Graphical representation and flow cytometry analysis results of expression of CD44 and CD24 surface markers in parental and 3D mammosphere cells: 92% 3D mammosphere cells are CD44+CD24-/low and 70% parental cells.

**Figure 4 ijms-21-01900-f004:**
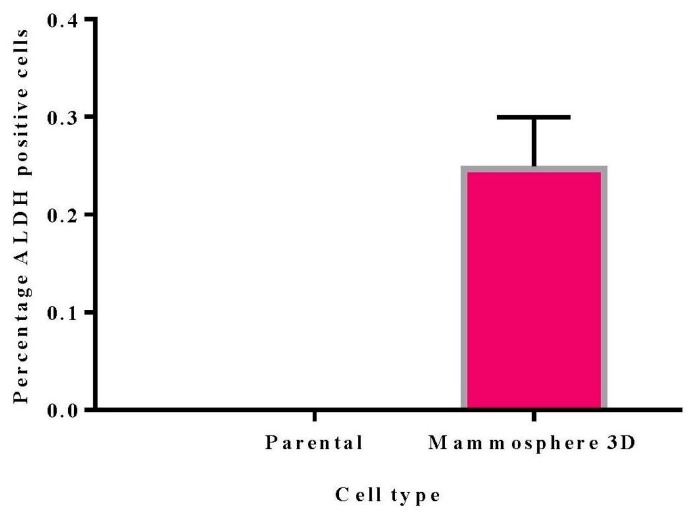
Graphical representation of ADLH activity in parental and 3D mammosphere cells.

**Figure 5 ijms-21-01900-f005:**
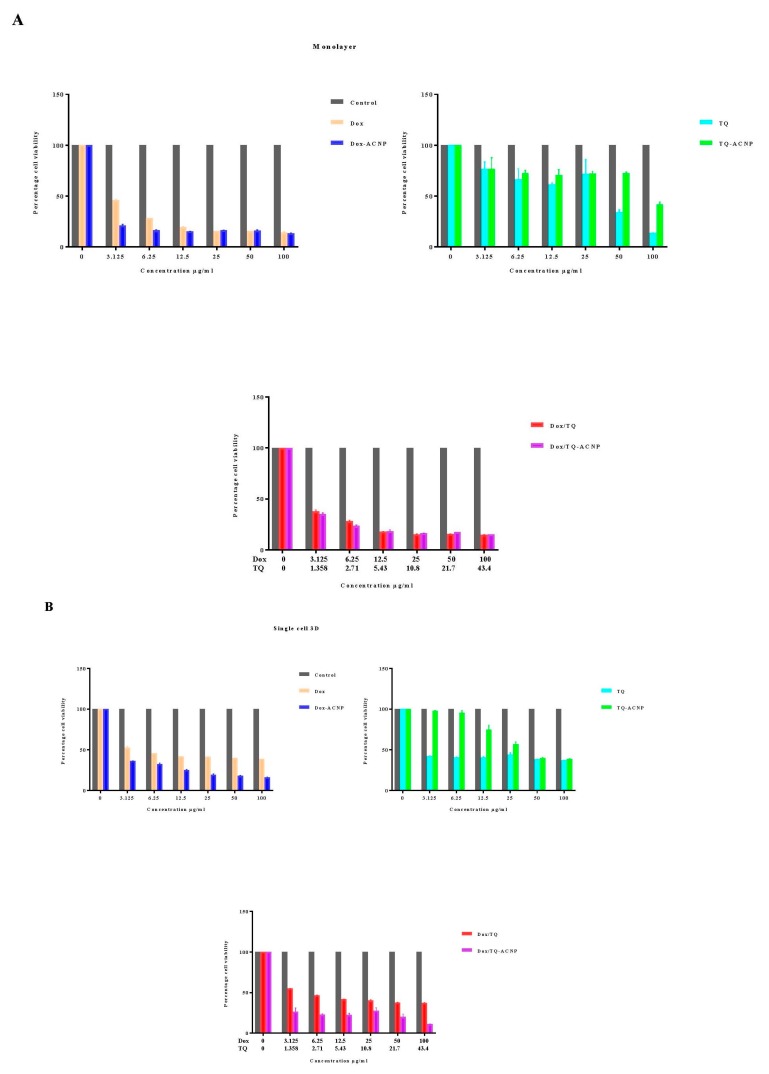
Graphs showing cell viability of (**A**) monolayer, (**B**) single cell 3D, and (**C**) 3D mammosphere after various treatments for 10 days.

**Figure 6 ijms-21-01900-f006:**
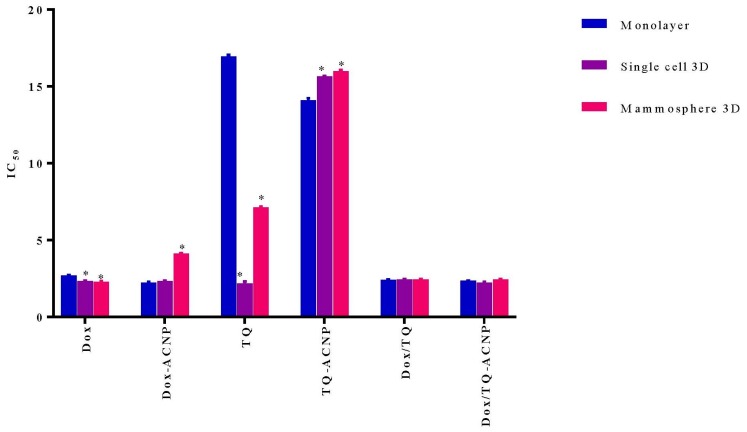
Graphical representation of IC50 data for the three culture conditions upon various treatments for 10 days. * *p* < 0.05 compared to monolayer.

**Figure 7 ijms-21-01900-f007:**
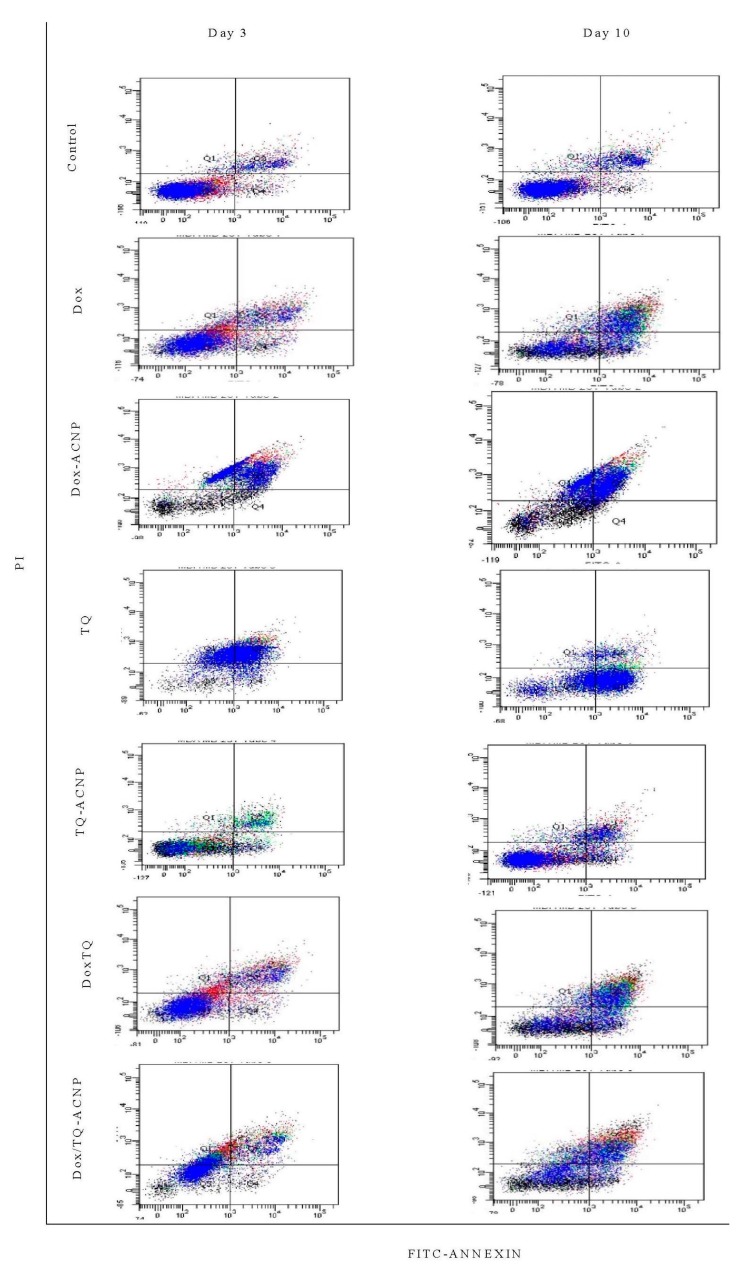
Flow cytometry results of cytopathology in control and treated 3D mammosphere cells; percentages of viable cells (Q3), early apoptosis (Q4), late apoptosis (Q2) and necrosis (Q1) at day 3 and day 10.

**Figure 8 ijms-21-01900-f008:**
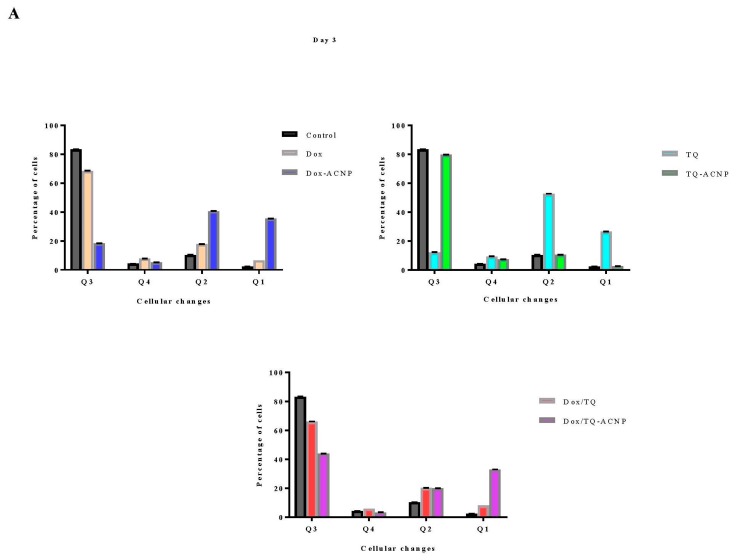
Estimation of percentage cytopathology in MDA MB 231 cells that were untreated and treated with free and drug-loaded nanoparticles. Percentages of viable cells (Q3), early apoptosis (Q4), late apoptosis (Q2) and necrosis (Q1) at day 3 (**A**) and day 10 (**B**).

**Figure 9 ijms-21-01900-f009:**
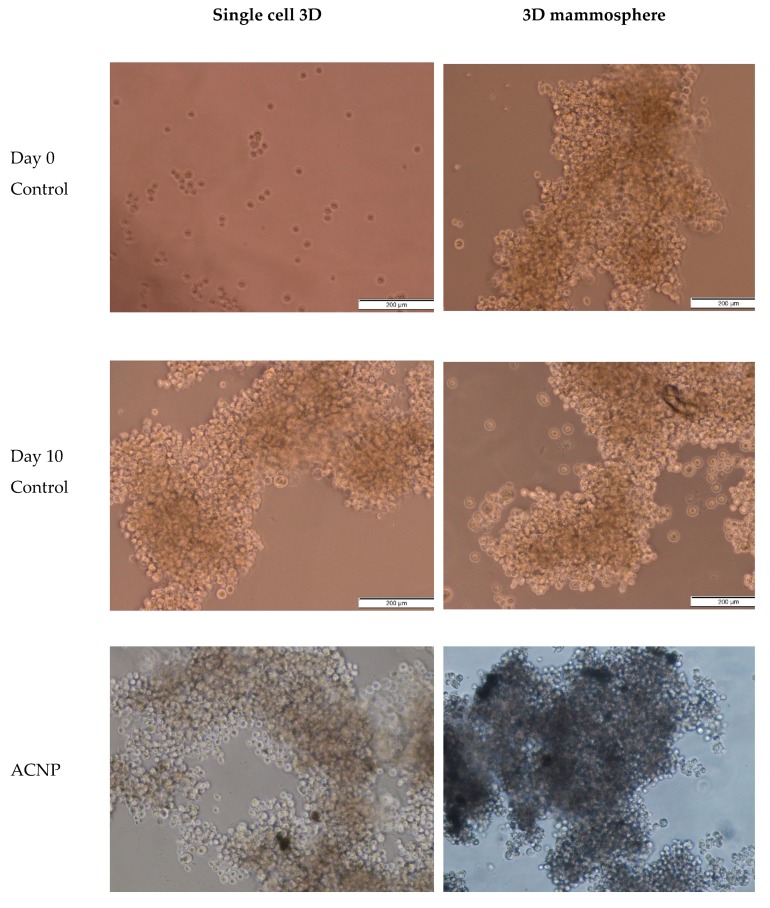
Photomicrographs of light microscopy showing the morphology of the mammosphere at days 0 and 10 after treatments.

**Figure 10 ijms-21-01900-f010:**
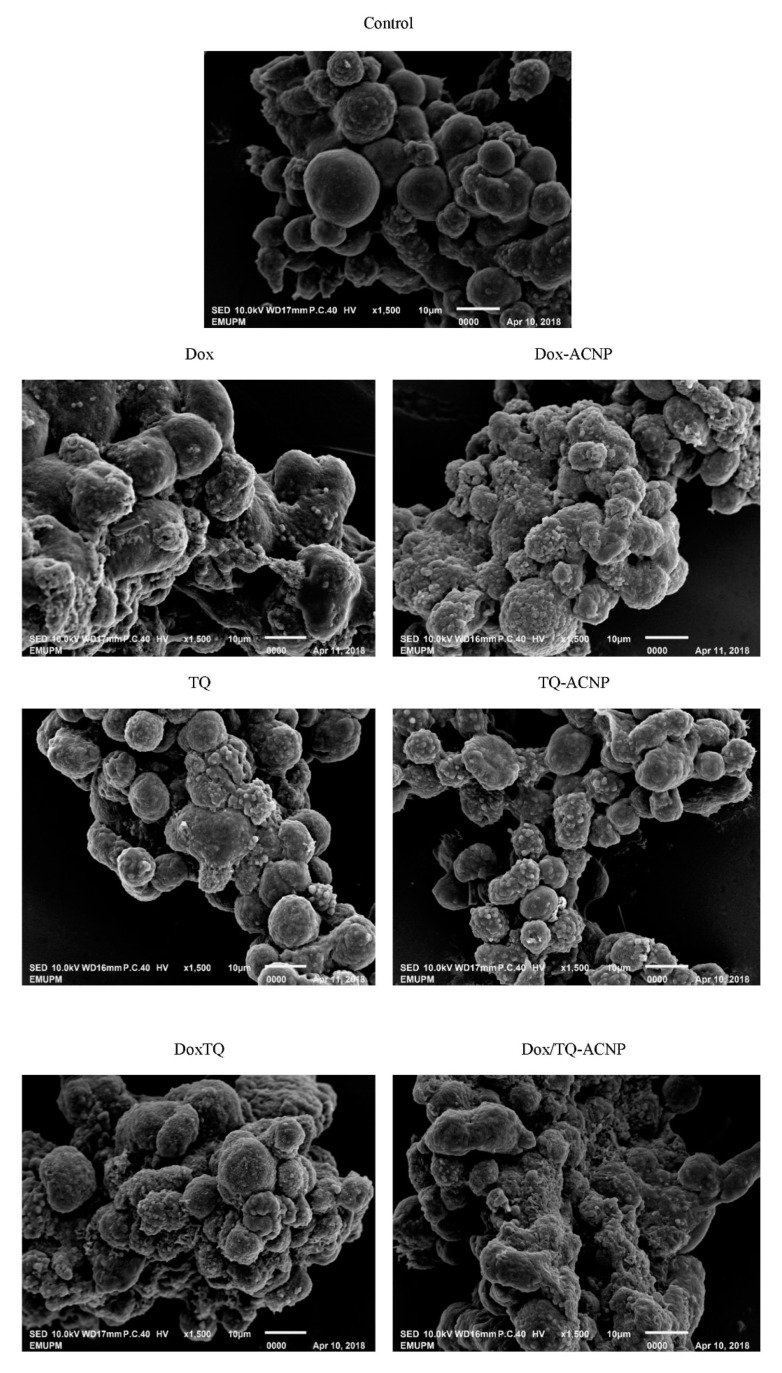
Scanning electron micrograph of 3D MDA MB 231 cells mammosphere after 10 days of treatment.

**Figure 11 ijms-21-01900-f011:**
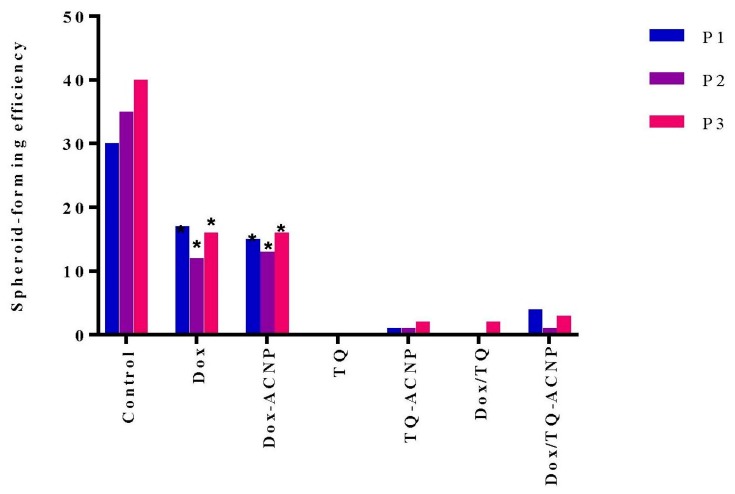
Self-renewal efficiency of MDA MB 231 CSCs after treatment at first passage through to the third passage. P1 = first passage; P2 = second passage; P3 = third passage. * *p* < 0.05 compared to control.

**Figure 12 ijms-21-01900-f012:**
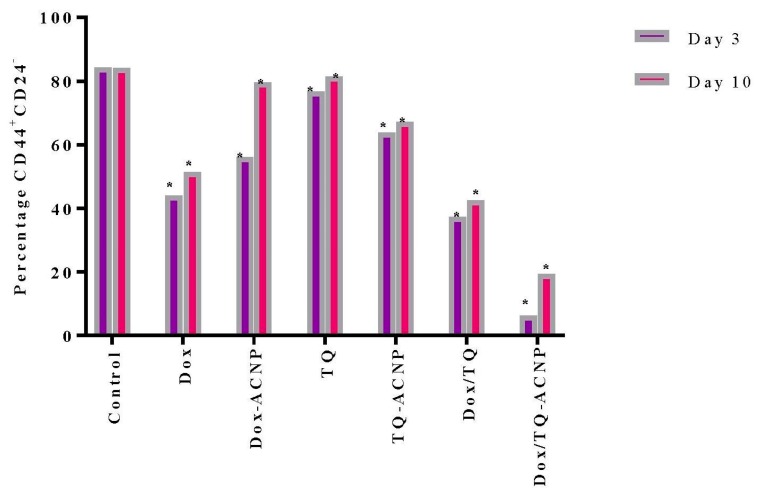
Graphical representation of CD44+CD24- cells after treatment at days 3 and 10. * *p* < 0.05 compared to control.

**Figure 13 ijms-21-01900-f013:**
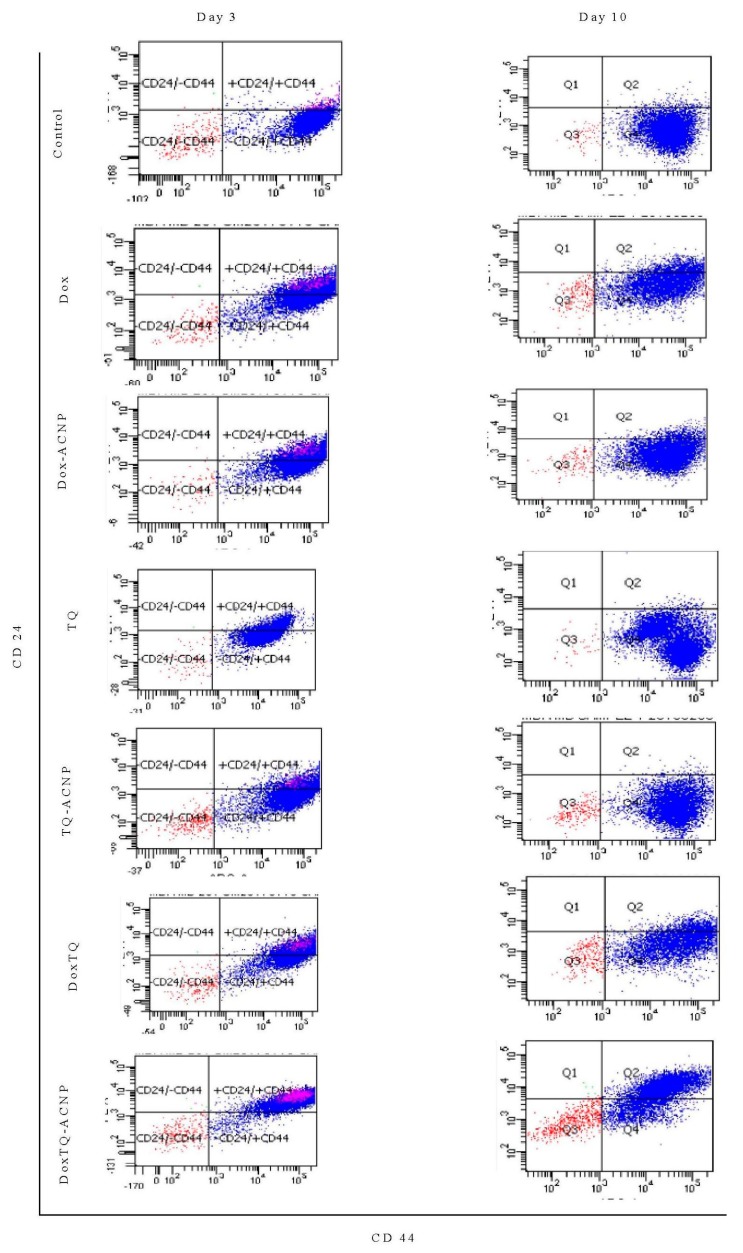
Flow cytometry representation of CD44+CD24- cells after treatment at days 3 and 10.

**Figure 14 ijms-21-01900-f014:**
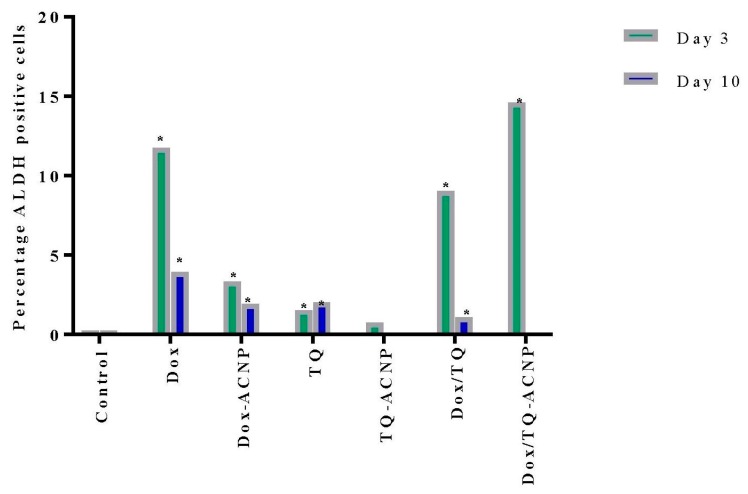
Graphical representation of ADLH activity after treatment at days 3 and 10. * *p* < 0.05 compared to control.

**Figure 15 ijms-21-01900-f015:**
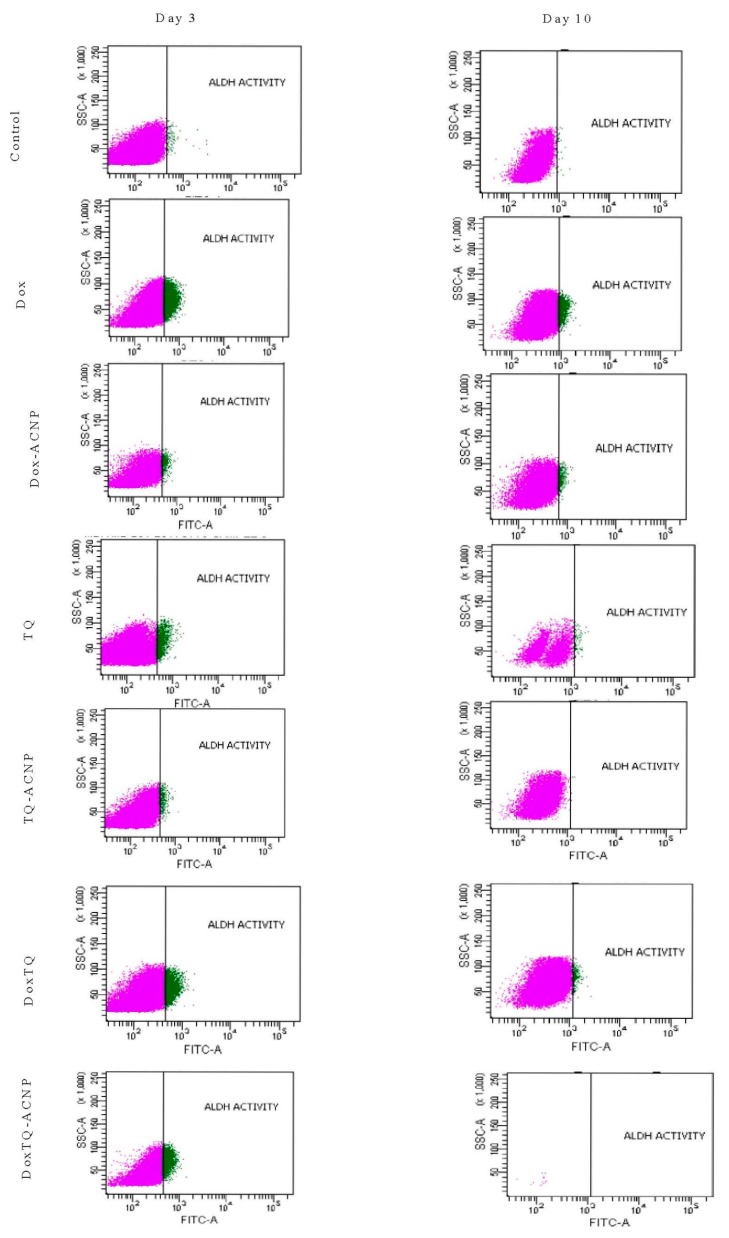
Flow cytometry result of ADLH activity after treatment at days 3 and 10. Purple and green indicate viable cells and ALDH activity, respectively.

**Figure 16 ijms-21-01900-f016:**
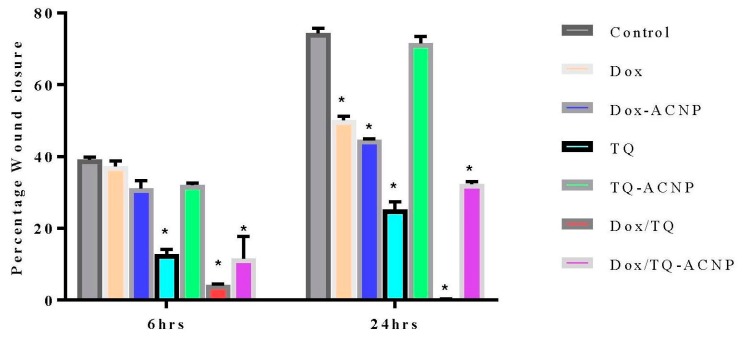
Graphical representation of wound closure in untreated and treated 3D mammosphere at 6 and 24 h post scratching. * *p* < 0.05 compared to control.

**Figure 17 ijms-21-01900-f017:**
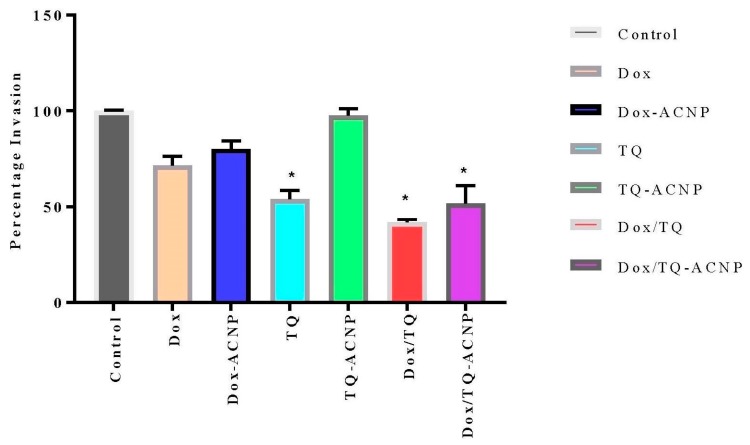
Graphical representation of the percentage of 3D mammosphere cell invasion across the basement membrane compared to control. * *p* < 0.05 compared to control.

**Figure 18 ijms-21-01900-f018:**
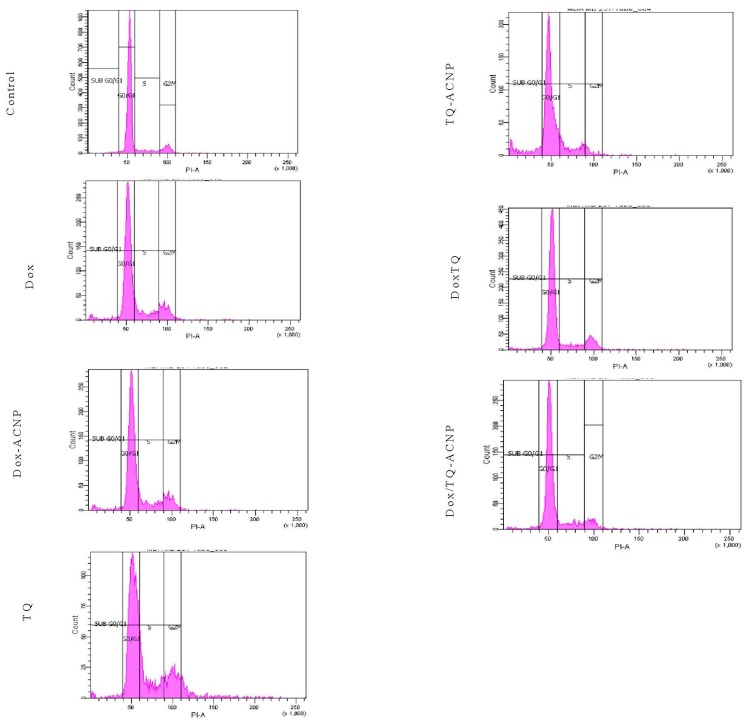
Flow cytometry data showing the effect of various treatments on the cell cycle distribution of 3D mammosphere cells at 48 h.

**Figure 19 ijms-21-01900-f019:**
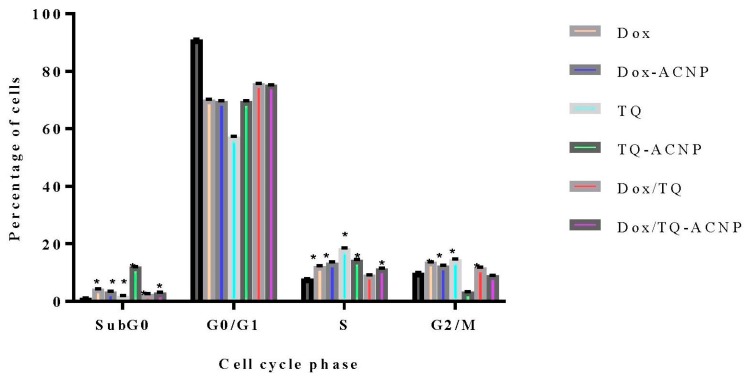
Graphical representation of the effect of various treatments on the cell cycle distribution of 3D mammosphere cells at 48 h. * *p* < 0.05 significant increase compared to control.

**Table 1 ijms-21-01900-t001:** IC_50_ data for the three culture conditions upon various treatment for 10 days.

	Dox (µg/mL)	Dox-ACNP (µg/mL)	TQ (µg/mL)	TQ-ACNP (µg/mL)	Dox/TQ (µg/mL)	Dox/TQ-ACNP (µg/mL)
Monolayer	2.705 ± 0.027	2.227 ± 0.122	16.95 ± 0.181	14.1 ± 0.239	2.418 ± 0.039	2.358 ± 0.039
Single cell 3D	2.339 * ± 0.043	2.342 ± 0.06	2.175 * ± 0.232	15.65 * ± 0.039	2.461 ± 0.041	2.224 ± 0.074
Mammosphere 3D	2.29 * ± 0.065	4.133 * ± 0.065	7.142 * ± 0.081	16.0 * ± 0.128	2.439 ± 0.048	2.446 ± 0.064

* *p* < 0.05 compared to monolayer.

**Table 2 ijms-21-01900-t002:** CI and interpretation for the free Dox and TQ combination treatment and Dox/TQ-ACNP against the 3D mammosphere.

	IC50	CI	Interpretation
Dox/TQ	2.439/1.06	1870	Antagonism
Dox/TQ-ACNP	2.446/1.063	0.12	Strong synergism

Interpretation of results: CI > 1.3 antagonism, CI 1.1–1.3 moderate antagonism, CI 0.9–1.1 additive effect, CI 0.8–0.9 slight synergism, CI 0.6–0.8, moderate synergism, CI 0.4–0.6 synergism; CI 0.2–0.4 strong synergism.

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
