# Peer review of "Cockle Shell-Derived Aragonite CaCO3 Nanoparticles for Co-Delivery of Doxorubicin and Thymoquinone Eliminates Cancer Stem Cells"

_ijms, 2020, doi:10.3390/ijms21051900_

Round 1
Reviewer 1 Report
The submitted manuscript investigated the use of ACNP combined with DOX or TQ, or DOX/TQ against MDA MD 231 breast cancer cell line. In general, I believe that this paper would be of interest to the readers of IJMS. However, additional discussion is necessary to describe what is shown by the data. As it currently stands, the manuscript has a little discussion of mechanisms that were actually studied. Please additional, more-specific comments below. 1. It would have been interesting to see if/how cell morphology changed after the ACNP treatment and why did Authors choose that nanostructures for drug delivery. 2. How the structure, morphology and physiochemical properties of ACNP including Zeta potential, FTIR spectrum has changed after functionalization of ACNP with DOX, TQ or DOX/TQ. 3. How the active surface of ACNP has changed after the functionalization by DOX/TQ or DOX, TQ. 4. Fluorescence/Confocal qualitative analysis could be beneficial for understanding the mechanism of action of ACNP combined with DOX/TQ 5. How did the nanoparticles pass the cell membrane and entered the cell, are they internalized via endocytosis? Does the size of the nanoparticles have any on influences their internalization efficacy? 6. It would have been also interesting to see how cell morphology changed from monolayer culture to mammospheres. 7. All Figures and graph should be improved and added at the higher resolution 8. The subtitle of the Figure should be improved. 9. In the Material and Method part, there was no information about the source of ACNP, drugs, and reagents like WST-1.
Author Response
We thank the reviewer for the comments.
1) The cellular morphology after treatment with ACNP have been described in our article titled “Ultrastructural Changes and Antitumor Effects of Doxorubicin/Thymoquinone-Loaded CaCO3Nanoparticles on Breast Cancer Cell Line” published in Frontiers in Oncology.
2-5) The structure, morphology the physiochemical properties, release profile, as well as the cellular uptake study of ACNP and the drug-loaded ACNP have been described in our article titled “Combine Drug Delivery of Thymoquinone-Doxorubicin by Cockle Shell-derived pH Sensitive Aragonite CaCO3 Nanoparticles” in Nanoscience & Nanotechnology-Asia. The article was referenced in the subsection 3.1 of Materials and Methods.
6) The cell morphology of monolayer culture after treatment has been described in our titled “Ultrastructural Changes and Antitumor Effects of Doxorubicin/Thymoquinone-Loaded CaCO3Nanoparticles on Breast Cancer Cell Line” published in Frontiers in Oncology.
7) All Figures and graphs have been improved.
9) The information on the source of drugs and WST-1 has been included in the manuscript.
Reviewer 2 Report
in the introduction, the similarity between breast cancer and the use of mammospheres could be commented a bit since this work uses these systems to carry out the study.
page 4 line 7. After seeing the recommended biblography it is not clear to me where the activity of the ALDH is derermined and that it is worth 0.6%.
In the studies of cell viability I find it coriors that viability obtained for the mixture, this values are very similar to those of the DOX alone. this fact can be explained form the mixing ratios, a question arises: why has the ratio 1:1 drugs not been studied?
page 7 table1. I believe that the exposed values require a revision of the significant digits and the values of the errors since these are only expressed with a digit unless this is a one.
page 12. In the photographs shown you cannot see the scale value correctly.
page 22 figure 18. The data shown in this graph I do not observe a quantifiable difference between the differnt treatment on the cell cycle distribution of 3D mammosphere cells at 48 hrs.
Some factors that can influence this study and should be commented more clearly: size and polydispersity of the nanoparticles, percent load of de drugos in the nanoparticles, the ionic force and pH.
Author Response
We thank the reviewer for the comments.
The Figure showing ALDH activities have been added.
The drug ratio that was used was the maximum ratio of thymoquinone and doxorubicin that could be loaded into ACNP. This was described in details in our article titled “Combine Drug Delivery of Thymoquinone-Doxorubicin by Cockle Shell-derived pH Sensitive Aragonite CaCO3 Nanoparticles” in Nanoscience & Nanotechnology-Asia. The article has been referenced in the manuscript.
Page 7, Table I: I do not understand the comment.
Figure 8 (page 12): A clearer image has been added.
Page 22, Figure 18. There was no significant difference in all the phases of cell cycle for all treatment. The significant shown were as compared to control.
The structure, morphology the physiochemical properties, release profile, as well as the cellular uptake study of ACNP and the drug-loaded ACNP have been described in our article titled “Combine Drug Delivery of Thymoquinone-Doxorubicin by Cockle Shell-derived pH Sensitive Aragonite CaCO3 Nanoparticles” in Nanoscience & Nanotechnology-Asia. The article was referenced in the subsection 3.1 of Materials and Methods.
Round 2
Reviewer 1 Report
This version of a manuscript titled "Cockle Shell-Derived Aragonite CaCO3 Nanoparticles for Co-Delivery of Doxorubicin and Thymoquinone Eliminates Cancer Stem Cells" had been improved and now it looks much better. All the comments and suggestions have been considered by the authors.
However, there are still some editing errors, like the quality of Figures legends or Figures subtitles. In my opinion, the authors should correct these errors in the manuscript.